# PEGylated graphene oxide elicits strong immunological responses despite surface passivation

Nana Luo[1,2,*], Jeffrey K. Weber[3,*], Shuang Wang[1,2], Binquan Luan[3], Hua Yue[1], Xiaobo Xi[1,2], Jing Du[4], Zaixing Yang[5], Wei Wei[1], Ruhong Zhou[3,5,6] & Guanghui Ma[1,2,7]

Engineered nanomaterials promise to transform medicine at the bio–nano interface. However, it is important to elucidate how synthetic nanomaterials interact with critical biological systems before such products can be safely utilized in humans. Past evidence suggests that polyethylene glycol-functionalized (PEGylated) nanomaterials are largely biocompatible and elicit less dramatic immune responses than their pristine counterparts. We here report results that contradict these findings. We find that PEGylated graphene oxide nanosheets (nGO-PEGs) stimulate potent cytokine responses in peritoneal macrophages, despite not being internalized. Atomistic molecular dynamics simulations support a mechanism by which nGO-PEGs preferentially adsorb onto and/or partially insert into cell membranes, thereby amplifying interactions with stimulatory surface receptors. Further experiments demonstrate that nGO-PEG indeed provokes cytokine secretion by enhancing integrin $\beta_8$-related signalling pathways. The present results inform that surface passivation does not always prevent immunological reactions to 2D nanomaterials but also suggest applications for PEGylated nanomaterials wherein immune stimulation is desired.

[1] State Key Laboratory of Biochemical Engineering, Institute of Process Engineering, Chinese Academy of Sciences, Beijing 100190, PR China. [2] University of Chinese Academy of Sciences, Beijing 100049, PR China. [3] Computational Biology Center, IBM Thomas J. Watson Research Center, Yorktown Heights, New York 10598, USA. [4] Institute of Biomechanics and Medical Engineering, Department of Engineering Mechanics, Tsinghua University, Beijing 100084, PR China. [5] Institute of Quantitative Biology and Medicine, SRMP and RAD-X, Collaborative Innovation Center of Radiation Medicine of Jiangsu Higher Education Institutions, Soochow University, Suzhou 215123, PR China. [6] Department of Chemistry, Columbia University, New York, New York 10027, USA. [7] Jiangsu National Synergetic Innovation Center for Advanced Materials (SICAM), Nanjing Tech University, Nanjing 211816, PR China. * These authors contributed equally to this work. Correspondence and requests for materials should be addressed to W.W. (email: weiwei@ipe.ac.cn) or to R.Z. (email: ruhongz@us.ibm.com) or to G.M. (email: ghma@ipe.ac.cn).

Bio–nano interfaces form when nanomaterials come into contact with biomolecular assemblies, such as protein complexes or lipid membranes[1,2]. Within a given medium, the physicochemical properties of bio–nano interfaces are mainly dictated by the diverse compositions, morphologies and surface chemistries that engineered nanomaterials can possess[3,4]. By tuning these characteristics, a myriad of nanomaterial functionalities can be realized for biomedical applications in biosensing, drug delivery, imaging and tissue engineering[5,6].

Important prerequisites for such biomedical applications involve establishing the *in vivo* stability and biocompatibility of nanomaterials in question[7,8]. Accumulating evidence has suggested that the intrinsic activities of nanomaterials are often overridden by the adsorption of biomolecular coronae from the biological milieu[9,10]. These coronal molecules bestow nanomaterials with new properties that transform their interactions at the bio–nano interface, interfering with both designed nanomaterial properties and innate biomolecular functions. To bypass these effects, nanomaterials can be coated with antifouling, hydrophilic and charge-neutral moieties such as polyethylene glycol (PEG) chains[11,12]. The resultant 'passivated' surfaces have been shown to discourage internalization by macrophages, allowing engineered nanomaterials to elude the body's preliminary line of defense against intruding particles[13–15]. Such characteristics, in principle, are thought to prevent macrophage activation and subsequent immunological response, thus ensuring the safe use of exogenous nanomaterials.

Two-dimensional (2D) nanomaterials have garnered particular attention due to their biomedical applicability[16–18]. Graphene derivatives, for example, possess large and specific surface areas that yield excellent adsorption propensies for drug delivery and have intrinsic photoluminescence properties that facilitate live cell imaging. Previous research into nanoparticle passivation, however, has largely concentrated on traditional spherical materials, such as micelles, liposomes and artificial polymers[19]. Accordingly, here we study the immunological impact of surface-passivated nano-graphene oxide (nGO), a prototypical and widely encountered 2D nanomaterial.

Intriguingly, we found that the macrophage response to PEGylated nGO was more dramatic than might be hypothesized. Despite our presumption of non-internalization being largely true, nGO-PEG was still shown to activate macrophages by promoting high levels of cytokine secretion. We discovered that this macrophage excitation was triggered through physical contact between nGO-PEG and cell membranes, interactions that enhanced cell mobility and migration. Applying gene chip analysis, we demonstrated that nGO-PEG stimuli were transduced into chemical signals through the upregulation of the integrin $\beta_8$ and activation of subsequent signalling pathways. Our molecular dynamics (MD) simulations support the notion that while PEGylated nanosheets are less likely to be internalized, they are even more likely to adsorb onto/partially insert into the membrane surface in face-on/edge-on configurations and thus solicit integrin-mediated signalling pathways. We explicate all of these results in depth below.

## Results

**Elevated cytokine response to nGO-PEG.** Foreign bodies that enter human serum are normally engulfed by macrophages, which in turn alter physiological behaviours involving cytokine secretion, inflammation and other related stress responses[20,21]. PEG is commonly conjugated to nanomaterial surfaces to avoid such internalization by immune cells[22,23]. Negligible internalization of nGO-PEG by macrophages was indeed observed in our experiments, as indicated by the absence of intracellular fluorescence signal (purple) in the nGO-PEG

confocal image (Fig. 1a). Signal from internalized nanosheets, however, is clearly present in cells exposed to pristine nGO. In further contrast with pristine nGO, which caused substantial nuclear damage to cells, the nuclear characteristics of nGO-PEG-exposed macrophages (for example, shape, area, roundness and intensity) remained consistent with those of normal cells (Supplementary Fig. 1). Coupled with the results of other viability tests (such as the CCK-8, Live-Dead and Annexin-V/PI assays) that can be found in our past work[13], our data suggest that nGO-PEGs are highly biocompatible.

In agreement with past observations, internalization of pristine nGOs stimulated the secretion of activation-associated cytokines such as interleukin (IL)-6, monocyte chemotactic protein-1, interferon-γ, tumour necrosis factor-α and IL-12 (Fig. 1b) in macrophages. But, contrary to our expectations, substantially greater cytokine production (even higher than in the positive control group of cells treated with lipopolysaccharides) was observed in macrophages incubated with the supposedly inert nGO-PEG (Fig. 1b). To explore these puzzling results, we first measured cytokine levels as a function of time (Fig. 1c). These data confirm that nGO-PEG indeed triggers the most conspicuous cytokine response among the materials tested (Fig. 1c). In a dosage-dependent assay (featuring nGO-PEG concentrations ranging from 2.5 to $10 \mu g \, ml^{-1}$), the levels of all activation-associated cytokines detected increased with increasing nGO-PEG dosage; little change was observed in the secretion of suppressive factor IL-10 (Fig. 1d). Neither free PEG nor a simple mixture of nGO and PEG (featuring no covalent conjugation between the two groups) aroused macrophage activity on the scale seen with nGO-PEG (Supplementary Fig. 2). Moreover, cytokine production levels were found to be positively correlated with the density of conjugated PEG chains (Supplementary Fig. 3). Chemical conjugation between PEG and nGO thus seems to be important for the marked increase in cytokine secretion observed with PEGylated GO.

**Impact of nGO-PEG on macrophage membranes.** We next sought to explicitly measure the impact of nGO-PEG on cell membrane parameters. After 24 h of incubation with nGO-PEG, confocal images revealed profoundly extended filopodia (green) intertwined with nGO-PEG (purple) on macrophage surfaces (Fig. 2a). These filopodia were also visible in transmission electron micrographs (Fig. 2b).

To gauge the impact of these nGO-PEG interactions on cell membrane integrity, we performed a lactate dehydrogenase leakage assay (Fig. 2c). Only very slight leakage was detected in the nGO-PEG-treated group, indicating that the membranes of nGO-PEG-exposed macrophages remain, in large part, intact and able to maintain normal cellular functions. As membrane motility is also highly correlated to membrane function, we labelled cell membranes with DiO and employed the fluorescence recovery after photobleaching technique to characterize membrane diffusion properties (Fig. 2d, Supplementary Fig. 4). According to the observed recovery kinetics, normal cells exhibited a recovery half time ($t_{1/2}$) of 12.21 s and a diffusion coefficient of $0.031 \mu m^2 s^{-1}$. By contrast, exposure to pristine nGO resulted in a recovery half time ($t_{1/2}$) of 7.51 s and an increased membrane diffusion coefficient of $0.145 \mu m^2 s^{-1}$. The slope of the fluorescence curve for nGO-PEG-treated cells was even steeper, featuring a half time of 5.5 s; the corresponding diffusion coefficient ($0.166 \mu m^2 s^{-1}$) was determined to be higher than that of either normal or nGO-exposed cells. This increased membrane mobility within the nGO- and nGO-PEG-treated groups is likely attributable to interactions between the nanosheets and cell membranes, a phenomenon that is more pronounced with nGO-PEG exposure. The precise mechanisms by which diffusive dynamics are

accelerated remain to be elucidated. As discussed in the context of our simulations, one expects direct contact with nGOs to freeze local membrane segments, arresting diffusion through site-specific interactions with lipids. Enhanced macrophage activation, however, could very well result in increased diffusion rates via some downstream process. Regardless of the underlying mechanism, heightened lipid mobility should serve to improve the transport properties of nGO-PEGs adsorbed onto the membrane, perhaps further amplifying macrophage activity.

The observed increase in membrane mobility could, in part, be associated with intensified macrophage migration induced by the presence of nGO-PEG (Fig. 2e). Within 1 h of observation,

normal cells persisted in quiescent and inactive states, remaining at their original locations. The addition of pristine nGOs caused cells to migrate only a small distance farther. Cells exposed to nGO-PEG, however, greatly extended their spheres of migration, with some cells even moving outside of our field of vision over the monitoring period (Supplementary Fig. 5). More quantitatively, nGO-PEG prompted cell trajectories to approach the periphery of a 30 μm spherical area, while untreated and nGO-treated cells remained near the centre of that region (Supplementary Fig. 6). The presence of non-internalized nGO-PEG thus is distinctly associated with an increase in macrophage motility, movement that is likely a result of their activation.

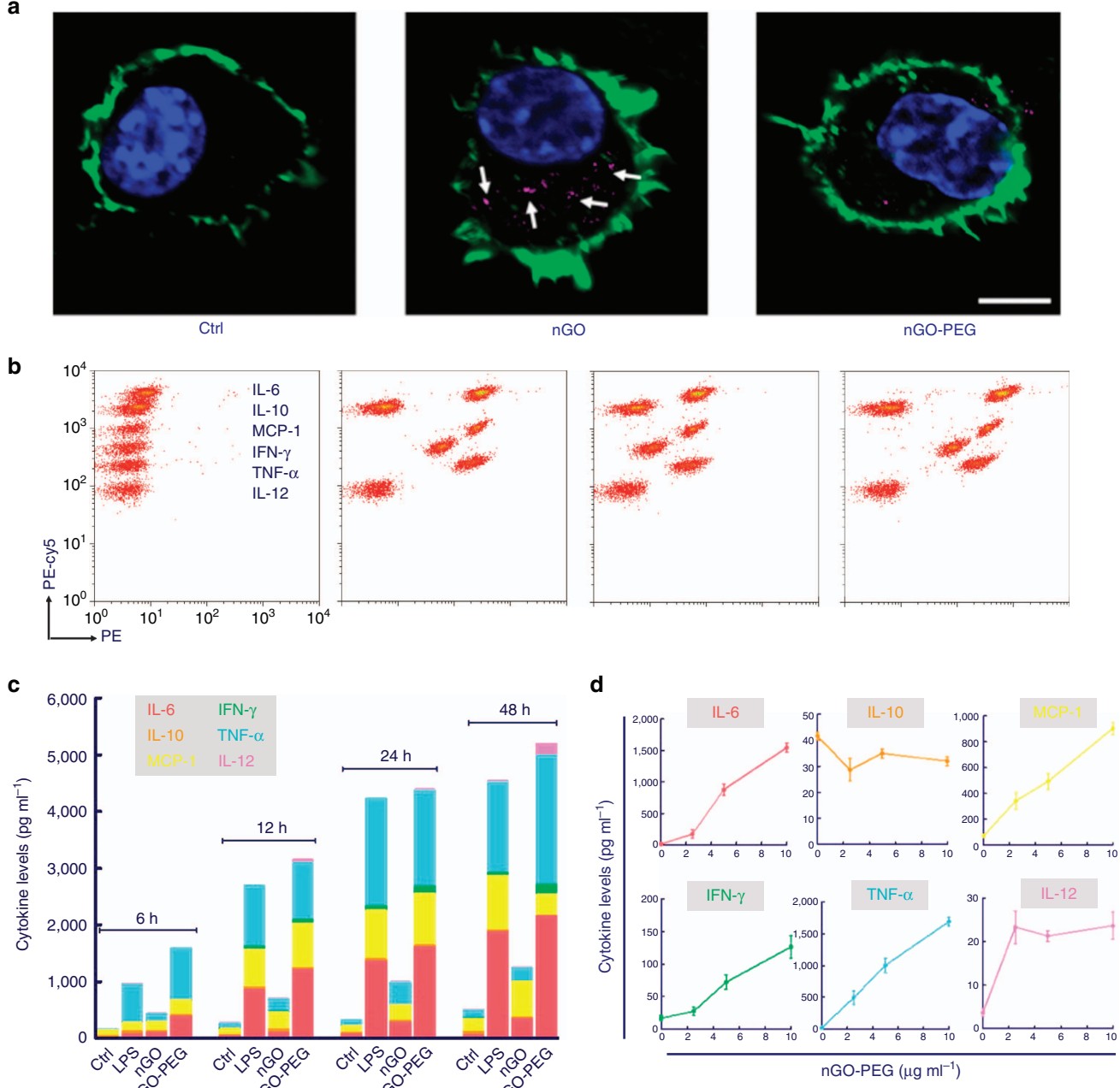

**Figure 1 | Impact of nGO and nGO-PEG on macrophage behaviour.** (**a**) Internalization of nGO and nGO-PEG observed by confocal imaging (purple dots, marked with white arrows: nGO complexes). Scale bar: 5 μm. (**b**) Flow cytometric dotplots of cytokine stimulation induced by nGO over 24 h. (**c**) Histogram of total cytokine concentrations as a function of time. Each cytokine concentration column is displayed as the mean value of three replicas. (**d**) Cytokine secretion induced by nGO-PEG at different concentrations after 24 h coincubation. Dosages in **a**–**c** were fixed at 10 μg ml⁻¹. Data are presented as means ± s.d., with $n = 3$.

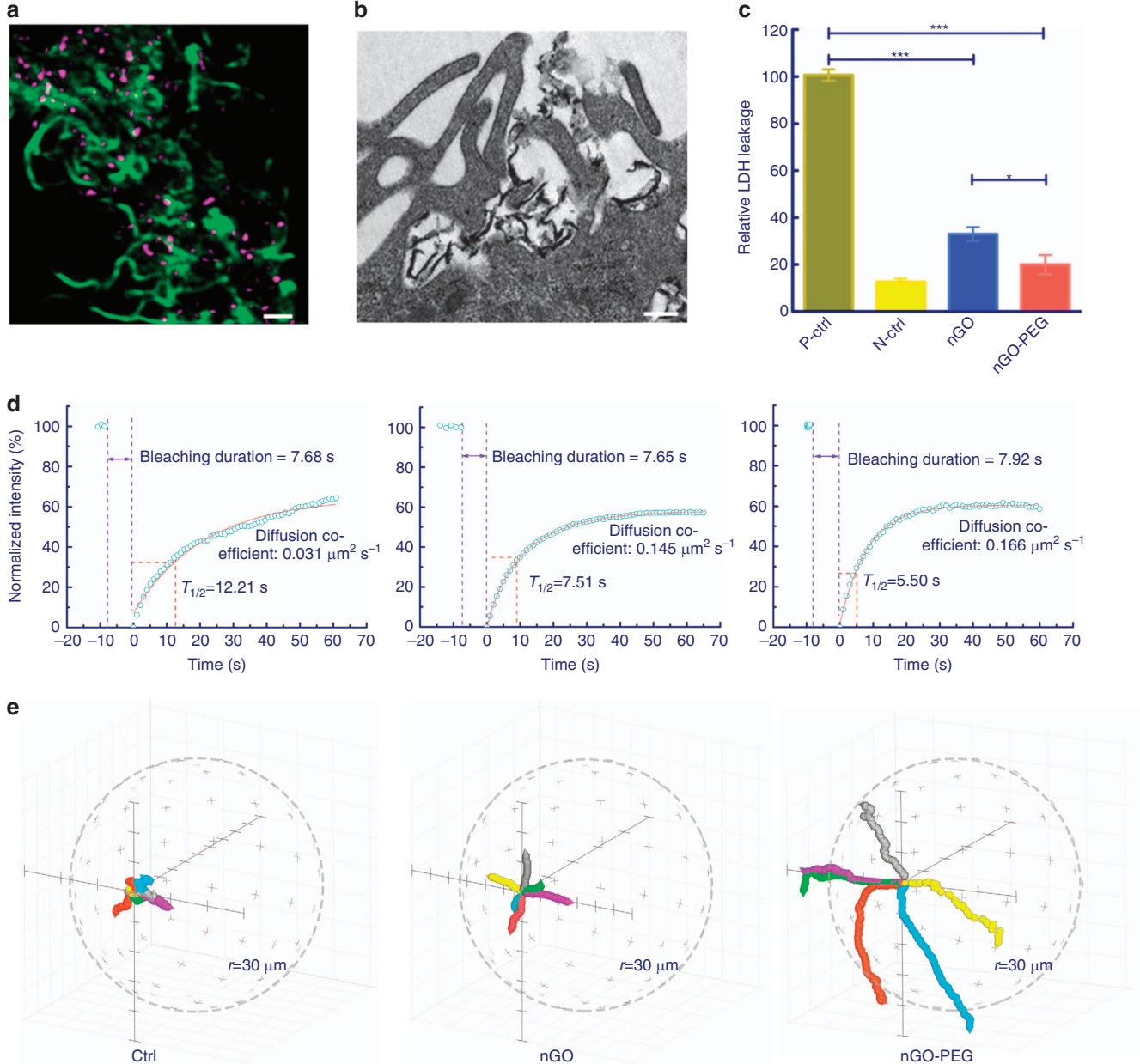

**Figure 2 | Impact of nGO and nGO-PEG on cell migration and membrane integrity.** (**a**) Confocal (top view; Scale bar: 1 μm) and (**b**) TEM images (side view; scale bar: 200 nm) showing interactions between nGO-PEG and macrophage filopodia. (**c**) Membrane integrity analysis conducted through lactate dehydrogenase leakage assays. The designations p-ctrl and n-ctrl represent positive and negative control, respectively. (**d**) Kinetics of macrophage membrane fluorescence recovery after photobleaching in the absence/presence of nGO and nGO-PEG. (**e**) Trajectories of cells in the absence or presence of nGO and nGO-PEG ($n = 6$ cells), where the graphical sphere radius is 30 μm (see Supplementary Movies 1 and 2 for more information). Data are presented as means ± s.d. with $n = 3$. *$P < 0.05$, ***$P < 0.001$.

**Molecular basis of membrane-nGO-PEG interactions.** Considering the unique properties possessed by 2D nanomaterials, one might posit that the planar structure of nGO-PEG could define the interactions responsible for its macrophage activation. As a rudimentary test of this hypothesis, we incubated macrophages in the presence of PEGylated carbon spheres (~200 nm) and one-dimensional carbon nanotubes (~4 μm length) (Fig. 3a). Under similar constraints on surface area and dose (10 μg ml⁻¹), 2D nGO-PEG induced, by far, the highest levels of cytokine secretion, membrane diffusion and cell migration (Fig. 3b, Supplementary Fig. 7). This trend in cytokine secretion is conserved among pristine nanomaterials as well (Supplementary Fig. 8). Based on simple physical arguments, 2D PEGylated

nanomaterials should indeed have the most pronounced interactions with cell membranes. The nGO-PEG, for example, might favour plane-to-plane interactions with cell membranes, which should cover more area (and thus support stronger van der Waals interactions) and persist longer than the point-to-plane and line-to-plane 'binding modes' of PEGylated carbon spheres and one-dimensional carbon nanotubes, respectively (Fig. 3c). Such planar interaction characteristics, in turn, could translate into high flux through activating receptors on the macrophage surface, as we discuss later.

Seeking clearer insight into the microscopic interactions between nGO-PEG and macrophage surfaces, we performed extensive molecular dynamics simulations of GO nanosheets

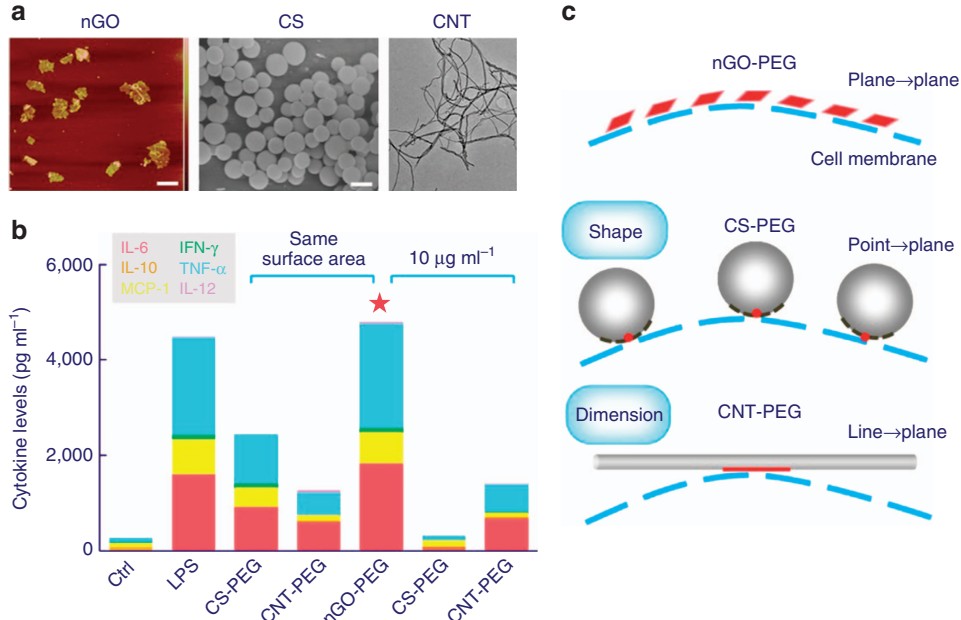

**Figure 3 | Effects of PEGylated carbon nanomaterials on cytokine secretion.** (**a**) Atomic force microscopic (AFM), scanning electron microscopic (SEM) and TEM images of graphene oxide, carbon spheres and carbon nanotubes, respectively. Scale bars: 200 nm. (**b**) Inflammatory cytokines secreted by macrophages when exposed to different PEGylated carbon nanomaterials, normalized by surface area and concentration. (**c**) Conjecture for interaction modes between carbon nanomaterials and cell membranes resulting in different levels of cytokine secretion.

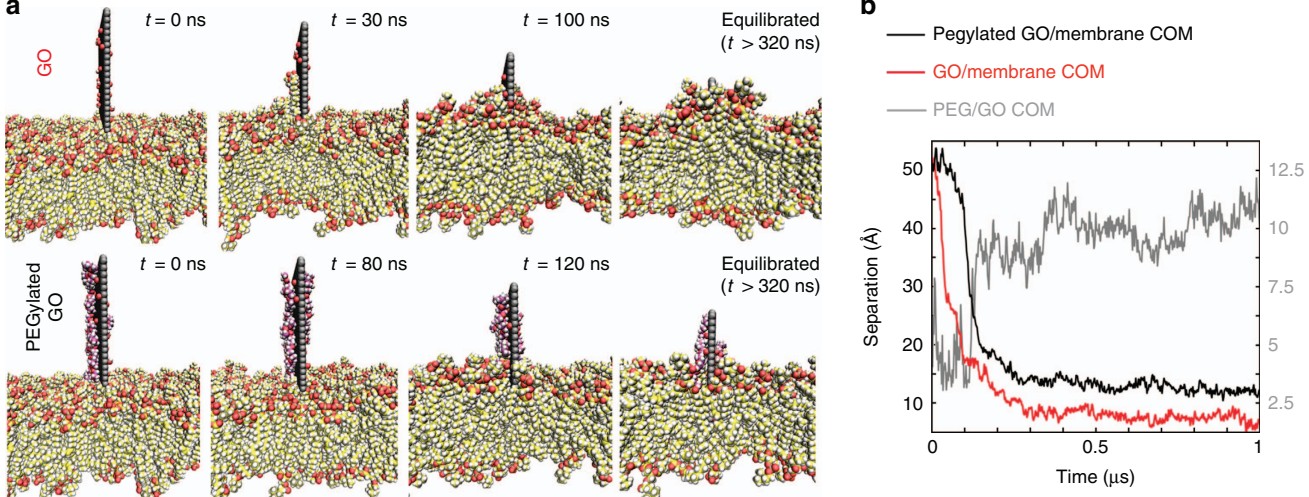

**Figure 4 | Simulation of nGO- and nGO-PEG-membrane interactions from edge-on configurations.** (**a**) System snapshots relevant to the observed membrane insertion processes, with GO carbons represented in grey and covalently linked PEG chains rendered in purple. (**b**) Centre-of-mass (COM) displacement data recorded over the course of the simulation trajectories. The PEG/GO COM separation trace (which highlights the PEG extrusion process during insertion) is presented on an alternative vertical scale.

(both pristine and PEGylated) in the presence of lipid membranes. Notably, past experimental and simulation work has demonstrated that pristine nGO has destructive effects on phospholipid bilayers via direct incisive mechanisms and further compromises membrane integrity through aggressive lipid extraction[24–28].

Corroborating this past evidence, both incisive and extractive mechanisms were indeed observed in our simulations of pristine nGO-membrane interactions (Fig. 4). Lipid extraction events buffeted the insertion process at early times (for example, at $t = 30$ ns), as recoil forces drew the nGO further toward the membrane centre. As the simulation proceeded, a depletion of lipid density became evident in membrane regions away from the point of nGO interaction. After a few hundred nanoseconds, the top of the nanosheet nearly disappeared below the membrane surface, almost penetrating the opposite side of the bilayer. Both this physical incision and the long-range density deficit induced by lipid adsorption onto the GO surface translate into destabilization of the membrane's structure; these results support past experimental observations of pristine nGO inflicting damage on macrophage cell membranes[13]. The complete nGO insertion observed here also suggests that nGO internalization by

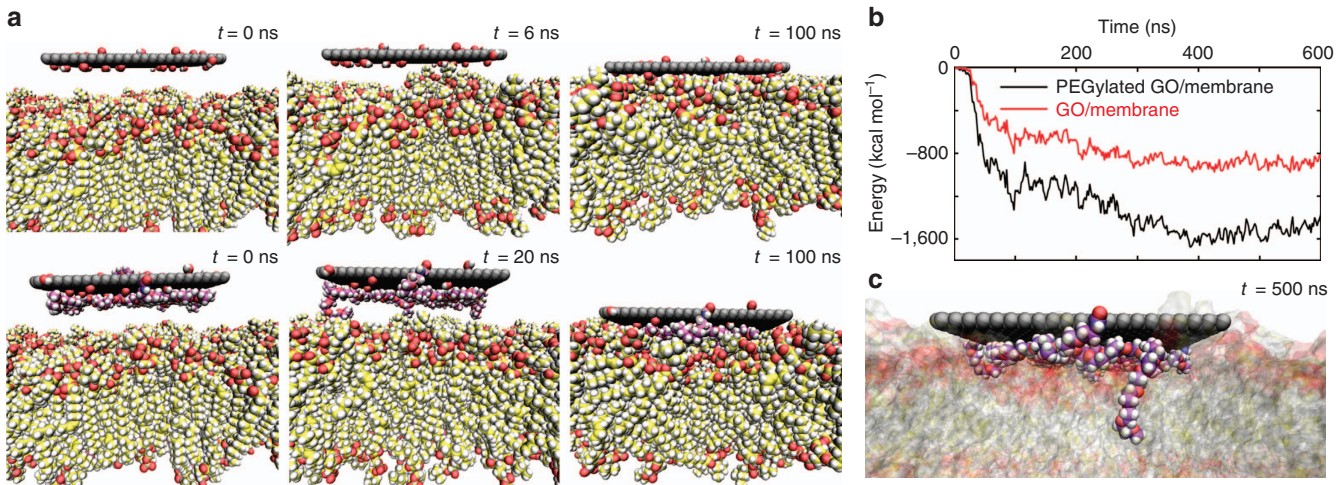

**Figure 5 | Simulation of nGO- and nGO-PEG-membrane interactions from side-on configurations.** (**a**) Representative snapshots of the membrane adsorption process. (**b**) Total interaction energies between pristine and PEGylated nGOs and the lipid bilayer. (**c**) Illustration of PEG desorption events that lead to the burial of PEG anchors in the membrane, events that further enhance nGO-PEG/membrane-binding energies.

macrophages might proceed via both passive and active means, as reported elsewhere[26,27,29].

Further MD simulations were performed to elucidate the passivating effects of PEG conjugation with nGO. A PEGylated GO nanosheet (with four PEG chains protruding from the nGO edges and two from the nGO faces) was first simulated in the absence of a membrane. Starting from fully extended polymer configurations, the PEG chains quickly collapsed and adsorbed onto the GO surface (Supplementary Fig. 9), as seen with protein adsorbates[30]. The 2D nature of nGO thus should largely be conserved in nGO-PEG. Placing an equilibrated nGO-PEG edge-on above the membrane surface, we observed that surface-bound PEG chains compete with lipid adsorption, impeding nanosheet insertion and preventing complete membrane penetration. Adsorbed serum proteins have been observed to mediate similar obstructions of graphene insertion[31]. As the centre-of-mass separation traces in Fig. 4b indicate, nGO-PEG insertion proceeds more gradually than in the case of pristine nGO. Interestingly, the bound PEG molecules resist desorption over the course of interactions with the membrane: the adsorbed chains are extruded upwards as lipids attach to the nGO surface and draw the nanosheet downward. Once available area on the GO surface has been exhausted, the insertion process terminates, leaving the PEG-covered graphene surface partially exposed on the membrane exterior. The damage incurred by the membrane, accordingly, is moderate compared with that seen for pristine nGO. Adsorbed PEG not only serves as a steric hindrance to membrane penetration but also occupies surface area that would otherwise attract lipids and further deplete the equilibrium density of the bilayer. One expects these basic consequences of PEG functionalization—that is, diminished membrane cutting and lipid extraction capabilities—to apply to nGO-PEGs of various sizes, explaining why PEGylated nanosheets are more benign than their unprotected counterparts[13].

The notion that nGO-PEG is less damaging to cell membranes may provide the key to understanding its immunoactive properties. Though macrophages should remain highly viable upon exposure to nGO-PEG, that viability does not prevent PEGylated nanosheets from adhering to cell membranes. Partially inserted nGO-PEGs, similar to that shown in Fig. 4, are likely to remain in that state for extended time periods and diffuse across the cell surface. It is even more likely, however, that nGO-PEG will also adsorb onto membranes in face-on configurations.

The simulations featured in Fig. 5 support this statement: face-on nGOs and nGO-PEGs immediately attached to the membrane surface and showed no signs of desorbing. Though estimating quantitative binding kinetics from these simulation trajectories is not possible, total interaction energies after face-on absorption favour the nGO-PEG by an approximate factor of two (Fig. 5b). An interesting phenomenon emerges that explains this surplus nGO-PEG interaction energy: after initial face-on contact, loops and termini of PEG chains desorb to form transient 'anchors' that bore into the lipid bilayer (Fig. 5c). One would not expect these single-chain anchors to compromise membrane integrity; however, the additional surface area and polar moieties made available by protruding PEG molecules enhance both electrostatic and van der Waals interactions with the membrane (Supplementary Fig. 10). Indeed, compared with pristine nGO, this augmented interaction energy is correlated with spatially tighter binding between nGO-PEG and the membrane (Supplementary Fig. 11).

Regardless of orientation, nGO-PEGs should thus be more capable than pristine nGOs of attaching to and diffusing across macrophage surfaces and activating cytokine-related receptors embedded in membranes. Our simulation data suggest several complementary reasons for the high activity of macrophages exposed to nGO-PEG. First, nGO-PEGs are unlikely to be internalized via direct/passive mechanisms, meaning adsorbed nGO-PEGs should persist longer on macrophage surfaces and elicit a correspondingly stronger cytokine response. Second, nGO-PEGs bound face-on to membranes seem more likely to remain in that state, as the substantially enhanced membrane interaction energies of nGO-PEGs indicate. Third, macrophages exposed to nGO-PEG are more likely to stay viable than those treated with pristine nGO, implying that the entire macrophage ensemble should be better able to transmit cytokine-related messages in response to nGO-PEG adsorption. Finally, adsorbed nGO-PEGs (present at a surplus compared with pristine nGOs) might also recruit immunoactive membrane proteins to nanosheet-binding sites by inducing changes in membrane curvature and dynamics; such effects have been proposed in previous work on graphenes and other classes of nanoparticles[29,32–34]. Due to the high computational cost of atomistic simulations, long-range curvature effects could not be probed in the present work. Though enhanced diffusion was observed upon nGO-PEG exposure at the cellular scale, one

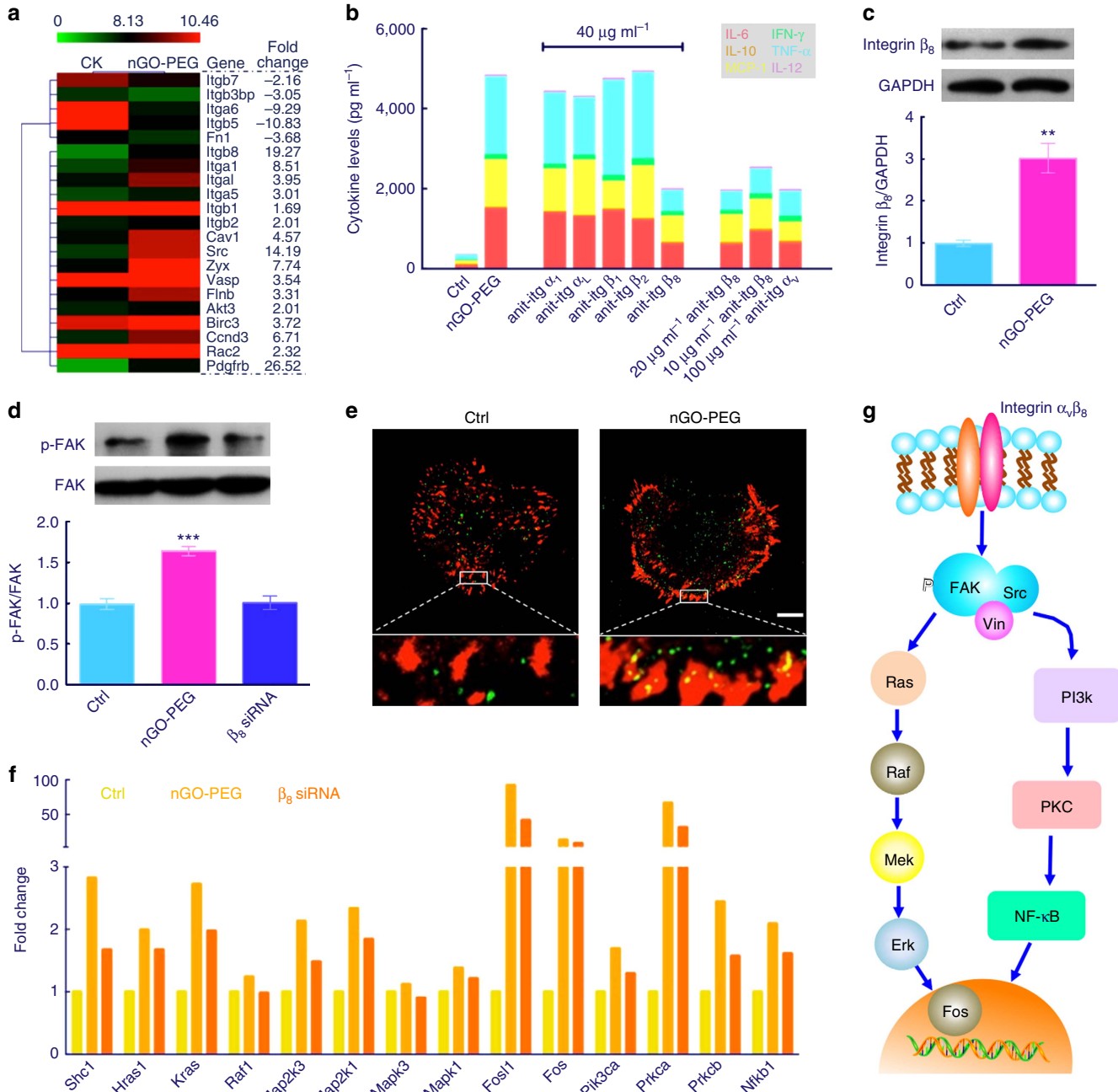

**Figure 6 | Mechanistic investigation of high cytokine secretion driven by nGO-PEG.** (**a**) Heatmap of membrane-related genes in the control and nGO-PEG groups. (**b**) Secreted inflammatory cytokine concentrations after the application of different integrin inhibitors or $\beta_8$ siRNA interference. (**c**,**d**) Western blotting analysis of integrin β8 and p-FAK after 24 h. Full gel images are included in Supplementary Figs 19 and 20. (**e**) Colocalization of integrin $\beta_8$ (green) and vinculin (red), with high-magnification insets included at the bottom. Scale bar: 5 μm.(**f**) Extent of gene expression change (measured by quantigene detection) in the control/nGO-PEG/siRNA groups after 24 h. (**g**) Proposed activation pathways that could result in high levels of cytokine secretion upon the introduction of nGO-PEG. Data are presented as means ± s.d. with $n = 3$. **$P < 0.01$ and ***$P < 0.001$.

expects an opposite effect in areas local to substrate/membrane contact. Direct lipid–nanosheet interactions should arrest lipid diffusion—via a process not unlike a glass transition—as our calculations confirm (Supplementary Fig. 12). Whether or not mesoscale diffusion and curvature effects emerge from basic membrane physics represents an intriguing topic for future study.

**Mechanism of nGO-PEG-induced cytokine secretion.** In order to alter the cellular behaviour, the above interactions between nGO-PEG and cell membranes must be converted into chemical signals[35,36]. To investigate the mechanism by which this signal transduction occurs, we performed a broad-spectrum gene screening on macrophages exposed to nGO-PEG. Based on our gene chip analysis, we noticed that a multitude of membrane protein-related genes were significantly upregulated and downregulated within the nGO-PEG group (Fig. 6a). Notable among translations of upregulated genes, integrins are known to be of particular importance for mediating cell adhesion, migration and the activation of divergent signalling pathways[37,38]. Using antibody-blocking experiments, we thus

explored the signalling pathways of the integrins most affected by nGO-PEG (Fig. 6b). Suppression of integrins $\alpha_1/\alpha_L/\beta_1/\beta_2$ had little effect on cytokine secretion upon exposure to nGO-PEG, but blockage of the most highly upregulated integrin $\beta_8$ (Fig. 6a,c), substantially reduced cytokine concentrations with near uniformity. As integrin receptors are composed of $\alpha$ and $\beta$ subunits, we also interrogated the behaviour of $\alpha_v$, the sole binding partner of $\beta_8$. In a manner complementary to $\beta_8$, a substantial (though slightly less dramatic) decrease in cytokine concentrations was also seen upon $\alpha_v$ blockage (Supplementary Fig. 13). This observation was conserved within the entire $\alpha_v$ subgroup, a set of proteins consisting of four additional heterodimers that were diluted to the effective concentration of $\alpha_v\beta_8$. A further short interfering RNA (siRNA) experiment (Supplementary Figs 14 and 15) demonstrated that a deficiency in $\beta_8$ gene expression diminished the effect of nGO-PEG on cytokine secretion (Fig. 6b), again suggesting a key role for $\beta_8$ in nGO-PEG-induced signal transduction.

Early in integrin signalling pathways, macromolecular complexes called focal adhesions serve as carriers that transmit signals into the cell interior[39]. This signalling process proceeds through the recruitment of focal adhesion kinase (FAK) to integrin clusters, which is activated upon binding via autophosphorylation[40,41]. Monitoring the expression of phospho-FAK (p-FAK) over the course of 2 days by western blotting (Fig. 6d, Supplementary Fig. 16), we found that nGO-PEG gradually provoked the production of p-FAK with a significant difference from normal cells ($P<0.001$). In addition to p-FAK, the protein vinculin is also a key component in focal adhesion complexes[42]. We thus imaged the distribution of integrin and vinculin using confocal microscopy (Fig. 6e). In contrast with untreated cells, the introduction of nGO-PEG not only promoted the expression of integrin $\beta_8$ but also affected its distribution around the cellular perimeter. Strong colocalization of vinculin and integrin (shown in the high-magnification inset of Fig. 6e) was also observed in response nGO-PEG exposure, suggesting that a proliferation of focal adhesions and concomitant signal transduction occurs as vinculin is recruited to nGO-PEG-binding sites.

Considering the putative pathways indicated by gene chip analysis (Supplementary Table 1), we made a further effort to quantify the expression of relevant genes through quantigene detection. As shown in Fig. 6f and Supplementary Figs 16 and 17, the presumed physical interactions with nGO-PEG result in a many fold upregulation of genes, such as Shc1, Kras, Map2k1 and Fos. Similar to our antibody-blocking experiments, pretreatment with integrin $\beta_8$ siRNA depressed the expression to levels more comparable to those of the control group.

Considered together, our results suggest a clear mechanism by which exposure to nGO-PEG stimulates cytokine secretion (Fig. 6g). Integrin $\alpha_v\beta_8$, residing on the membrane surface, is activated by interactions with membrane-adsorbed or -inserted nGO-PEG, resulting in vinculin recruitment and autophosphorylation of FAK. P-FAK then arouses elevated expression of the genes Ras, Raf, Mek and Erk, which in turn governs the enhanced expression of the nuclear transcription factor Fos. The genes for phosphoinositide-3 kinase and protein kinase C are also activated, encouraging the translocation of nuclear factor-κB from the cytoplasm into the nucleus (Supplementary Fig. 18). These events act in concert to boost cytokine synthesis and secretion by macrophages, potentially leading to further immunological responses downstream.

In conclusion, surface passivation is widely considered to be an effective dampener of crosstalk at the bio–nano interface, improving nanomaterial stability and biocompatibility and circumventing macrophage internalization and activation.

However, our studies have demonstrated that PEGylation of 2D nanomaterials is less passivizing than previously believed. Although stable, biocompatible and non-internalized, nGO-PEG was still able to activate macrophages by triggering a potent release of cytokines. One suspects nGO-PEG's high available surface area for membrane interactions—representative of all 2D nanomaterials—dictates stimulatory effects on the cells in question. Distinct changes in membrane morphologies were observed upon nGO-PEG exposure, alongside augmented membrane mobilities and enhanced cell migration. Our simulation work on nGO-PEG/membrane interactions suggested specific molecular mechanisms (via edge- and face-on contact) by which macrophage activation might be facilitated. Further experiments established that the integrin $\alpha_v\beta_8$ plays a crucial initiating role in signal transduction related to nGO-PEG/membrane binding. Through the upregulation of integrin $\beta_8$ and the subsequent activation of FAK-related intracellular signalling pathways, the external stimulus carried by nGO-PEG is transduced into chemical signals that ultimately give rise to macrophage activation.

Importantly, the present study indicates that surface passivation might not always allow 2D nanomaterials to escape immunological responses. We did note an occurrence of cytokine downturn at the time of second stimulus (Supplementary Fig. 21), an observation that suggests nGO-PEG-induced macrophage activation might decay over time. Interestingly, very comparable levels of activation were also observed with the PEGylation of the non-carbon-based 2D nanomaterial $MoS_2$ (Supplementary Fig. 22). Questions as to whether such activation by 2D PEGylated nanomaterials would prime further inflammatory or immunological responses, and to what extent, demand further *in vivo* experimentation. In the meantime, it would be worthwhile to explore the potential for nGO-PEG-induced activation in related systems such as dendritic cells, which are critical antigen-presenting cells in the immune system[43]. Indeed, our observations also suggest that PEGylated nanomaterials may be suitable for use in situations in which the immune system requires stimulation[44,45]. The fact that nGO-PEG elicits a vigorous immune response without causing conspicuous damage to macrophages is perhaps promising from a therapeutic perspective. Though more research is certainly needed, targeted cytokine secretion induced by carefully delivered nanomaterials such as nGO-PEG could serve as an effective component of future immunotherapies. Further work on the immunological implications of nGO-PEG exposure will not only supplement our current knowledge of the effects of surface functionalization but will also pave the way for safer and more effective biomedical applications for these novel nanomaterials.

## Methods

**Cell cultures.** Peritoneal macrophages (PMØs) were harvested from stimulated C57BL/6 mice according to a typical protocol[46] and cultured under standard conditions. C57BL/6 mice were ordered from Charles River Laboratories (USA). All animal experiments were performed in compliance with the institutional ethics committee regulations and guidelines on animal welfare. The murine macrophage cell line J774A.1 was supplied by the ATCC (American Type Culture Collection). More details of cell culture and other methods can be found in Supplementary Methods.

**Nanomaterial synthesis and characterization.** PEGylation of pristine nGO (single layered, with ~200 nm lateral size), as well as other carbon-based nanomaterials, was performed based on previously established methods[47]. In brief, 1-Ethyl-3-(3-dimethylaminopropyl)carbodiimide (EDC) (20 mM) was introduced into a pristine nGO suspension (~500 µg ml$^{-1}$) and sonicated for 15 min, after which mPEG-NH$_2$ was added and allowed to react overnight. The final products were harvested by centrifugation at 70,000$g$ after repeated washing with deionised water. Nanomaterial morphologies were imaged using an atomic force microscope (Bruker), a scanning electron microscope and a transmission electron microscope (JEOL). More detailed characterizations of nGO and nGO-PEG are included in

Supplementary Figs 23 and 24; Supplementary Tables 2 and 3 and our previous study[13].

**Cytokine secretion measurements.** To measure cytokine secretion levels, PMØ cells were exposed to various GO solutions over distinct time periods (6, 12, 24 and 48 h) and at different dosages (10 and 40 µg ml$^{-1}$). Cytokine secretion (IL-6, IL-10, IL-12, tumour necrosis factor-α, monocyte chemotactic protein-1, and interferon-γ) was evaluated by flow cytometry (Beckman Coulter).

**Cell imaging.** For imaging applications, PMØs were seeded ($1 \times 10^5$ ml$^{-1}$) in a petri dish and incubated with nGO complexes at 10–40 µg ml$^{-1}$ for 24 h. NGO complex imaging was performed using flow cytometry or confocal laser scanning microscopy (Leica), invoking graphene's intrinsic photoluminescence. The cytoskeleton and nuclei were separately stained with rhodamine-phalloidin (green pseudocolour in images, Invitrogen) and Hoechst. For characterization of membrane morphologies, cells treated with nGO-PEG were stained and fixed, cut on a Reichert Ultracut microtome (Leica) and imaged using transmission electron microscope. Lactate dehydrogenase leakage assays and fluorescence recovery after photobleaching experiments were conducted using standard procedures. Cell trajectory videos were recorded inside an Ultraview (PerkinElmer) incubator in bright field mode.

**Gene chip analysis.** In preparation for gene chip analysis, PMØ cells were cultured and then exposed to 10 µg ml$^{-1}$ nGO-PEG for 12, 24 and 48 h, respectively. RNA was extracted from each sample using 0.6 ml of trizol reagent and sent to Shanghai Gene Corporation for further analysis. Pathway analysis was performed using the KEGG database. Quantigene detection was carried out on PMØ cells split into control, nGO-PEG and β$_8$-gene-silenced nGO-PEG groups after 12, 24 and 48 h of coincubation.

**Antibody-blocking experiments.** Anti-integrins β$_1$, β$_2$ (Millipore), α$_1$ (Abcam), α$_L$, β$_8$ (Santa Cruz Biotechnology) and α$_v$ (Biolegend) were added at the indicated concentrations and allowed to incubate 4 h; cells were subsequently exposed to culture medium containing 10 µg ml$^{-1}$ nGO-PEG for 24 h. Cytokine levels were then detected as described above. Transfection of mouse integrin β$_8$ siRNA into PMØs was performed with an RNAimax reagent (Invitrogen) according to the manufacturer's instructions.

**Western blotting.** Primary antibodies for GADPH (1:1,000, Goodhere Corporation), FAK (1:1,000, Santa Cruz), p-FAK (1:500, Cell Signaling Technology), integrin β8 (1:500, Biolegend) and horseradish peroxidase-conjugated anti-rabbit IgG (1:2,000, Cell Signaling Technology) were used for immunoblotting analysis. Immunofluorescence experiments were conducted using β$_8$ integrin (1:200) and vinculin (1:200, Sigma-Aldrich) for primary antibodies and anti-rabbit IgG conjugated with Alexa Fluor 488 and anti-mouse IgG with Alexa Fluor 647 dyes for confocal laser scanning microscopic imaging. Quantigene detection was carried out on PMØ cells split into control, nGO-PEG and β$_8$-gene-silenced nGO-PEG groups after 12, 24 and 48 h of coincubation.

**Simulation parameters and configuration.** nGO sheet coordinates and chemistries were first generated (using the VMD Nanotube Builder plugin[48]) to comply with a slightly reduced version of the standard Lerf–Klinowski model[49]. Starting PEG chain configurations, each consisting of 15 monomers, were then created manually; PEG chains were covalently attached to nGO through amide linkages similar to those used in our experiments. Force field parameters for PEG monomers and termini were extracted directly from the CHARMM ether force field[50], while parameters related to GO functionalities and amide linkages were adapted from similar motifs within the CHARMM27 force field[51].

**Simulation setup.** The initial nGO-PEG configuration was minimized and equilibrated in isolation using the NAMD simulation package[52], applying a Langevin integrator (310 K; 1 atm), the CHARMM27 force field, the TIP3P water model, PME electrostatics and normal SETTLE constraints. An equilibrated snapshot of our nGO-PEG system is shown in Supplementary Fig. 10. The 1-palmitoyl-2-oleoyl-sn-glycero-3-phosphocholine (POPC) membrane fragments featured in our simulations were generated and equilibrated using a standard procedure.

**Production simulations.** For production simulations, an equilibrated nGO-PEG (or pristine nGO) was initialized above the membrane in either a face- or edge-on configuration; trajectory data were collected for several hundreds of nanoseconds (in some cases, for a microsecond) using the same force field and simulation parameters described above.

**Data availability.** Data supporting the findings of this study are available within this article (and its Supplementary Information file) and from the corresponding author upon reasonable request.

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

## Acknowledgements

We thank Ding Ma's Group of Peking University for offering the use of GO and CNT powders; we also thank Hua Zhang's group of Nanyang Technological University for the use of MoS$_2$ solution. This work was partly supported by National Science and Technology Major Project (2014ZX09102045-004), 973 Program (2013CB531500) and National Natural Science Foundation of China (21320102003, 11374221, 11574224, 31370018). This project was partially funded by the Priority Academic Program Development of Jiangsu Higher Education Institutions (PAPD). R.Z. acknowledges the support from the IBM Blue Gene Science Program (W125859, W1464125, W1464164). J.D. is also supported by Tsinghua University (2012Z02133).

## Author contributions

G.M. and R.Z. conceived the study and designed all experiments. N.L., S.W., H.Y., X.X. J.D. and W.W. performed and analysed all experiments. J.K.W., B.L., Z.Y. performed the molecular dynamics simulations and computational analysis. N.L., J.K.W., W.W., R.Z. and G.M. wrote the manuscript with support from all authors.

## Additional information

**Competing financial interests:** The authors declare no competing financial interests.

**Publisher's note**: 

