## [Peer Review File · Nature Communications]

Reviewer #1 (Remarks to the Author):

The authors in this article present some new evidences about the role of PEG functionalisation of GO as a passivation method in biomedical applications. In contrary with what is believed up to now, it is showed that pegylation of GO induces the highest level of cytokine secretion, which means analogous macrophage activation. The article is well documented and written and present very important and novel results and I would suggest the publication to 'Nature Communication' after a minor revision according to the following comments.

Below some comments related the manuscript.

- a) The phrase in line 103 seems to be in contrast with Fig. 1a. (absence of intercellular fluorescence).
- b) The phrase in line 150, 'becomes more hydrophobic after the nGO-PEGs adsorption'; how this is happen since PEG is highly hydrophilic ?
- c) The authors should mention the origin and the characteristics of GO (single layers, few layer nanosheets etc). The size of GO nanosheets is very important in bioapplication and it is not clearly referred in the article. Even if we suppose that the mean size of GO is 200 nm (as the bare in the TEM image) it would be very interesting to study GO-PEG with smaller size e.g 50 nm.

Reviewer #2 (Remarks to the Author):

The manuscript entitled "PEGylated Graphene Oxide Elicits Strong Immunological Response Despite Surface Passivation" describes very detailed insight into the interaction between 2D material and the biological interface (macrophage surface). The detailed molecular biology of an effect of PEGylated graphene oxides on macrophages is further supported by molecular dynamic simulations of interactions between the biological membrane and the tested material. However, the Nature family journals require very precise and detailed work in all areas related to the manuscript. The present story totally fails in materials characterization. The basic and standard microscopic (TEM/SEM) and spectroscopic analyses (IR/Raman/AAS/XPS etc.) are missing. This fact disables to understand/explain objectively the phenomena behind the triggering of immunological response of PEGylated GO. For this reason, I recommend to reject the manuscript in its current form.

I suggest the authors to address some issues for further submission(s).

*Zeta potential measurements for tested nGO and nGO-PEG are missing. The surface charge is an important parameter affecting the interactions of nanomaterials on the nano-bio interface. Is there any change of this parameter in the mixture of nGO/PEG which was tested for cytokine secretion?

*The given AFM of nGO-PEG does not contain the scale bar, similarly as for the carbon particles. A more detailed microscopic characterization should be added, e.g. the difference in the AFM height-profiles of bare nGO and nGO-PEG can support the theoretical model of nGO-PEG. The thickness of PEG shell would also be quantifiable.

*The authors show the cytokine secretion as the function of surface area of carbon particles, carbon nanotubes and graphene oxide sheets? How did they measure this parameter (surface area)? This information should be given in ESI.

* The effectivity of the PEG attachment (the total coverage) of graphene oxide sheets has to be determined because it is the crucial point in further biological experiments. For example, XPS analysis with determination of nitrogen (from NH₂-PEG) should give information on the amount of PEG chains per particle. Similarly, a TG analysis should be involved.

*Some quantification of metal impurities should be added to exclude their possible role in the biological response (typically AAS).

*In the confocal images, the nGO-PEG should be monitored by purple/violet color. Does the sample exhibit any fluorescence? Why the fluorescent spectra are not involved? What is the quantum yield? Hence, the signal from nGO is not visible in Fig 1a /the second picture. If there is any signal, the authors should mark it by arrows.

*In the simulation setup, harmonic restraint should be described in more details, e.g. how many atoms are restrained (the force constant of restraint etc.).

*Simulations of edge-on nGO with or without PEG are not completely equilibrated - there is still a trend in Figure 4.

*Discussed lipid extraction or effect on membrane curvature was not observed in any simulation?? Also the changes in the dynamics of the membrane should be calculated from the simulations (at least for the equilibrated side-on simulations) prior to any discussion of the significance of the computation data.

Without all these data, it is difficult to extract the key message of the manuscript! The role of PEG would be due to so many reasons and the authors should identify the key parameters precisely. Moreover, it should be clearly proved if the PEG effect in stimulation of strong immunological response is generalizable also for other nanomaterials; still, the present data do not prove it.

Reviewer #3 (Remarks to the Author):

The present manuscript, entitled "PEGylated Graphene Oxide Elicits Strong Immunological Responses Despite Surface Passivation", demonstrates that 2D nano-particles can elicit enhanced inflammatory responses even in the presence of an "anti-fouling" coating.

The data and results appear to be novel, and interesting; however, the authors should take the time to more fully explain their results in the context of what is known in the literature. The authors find an unexpected result, but do not do a sufficient job of explaining the potential differences in their findings. Additionally, there appear to be many controls missing which would have made the claims more convincing.

There is significant data to support the authors claims; however, some figures are difficult to interpret. The use of bright colors in the figures makes some of them hard to read. Yellow text, for example is not easily distinguished. In figure 6e, the authors state the colocalization of green and red, I cannot see any green in these panels.

I have questions about some of the methodology used and how the results are interpreted. In figure 1, the authors state that nGO are internalized, but I cannot see any evidence of this in your figure. The authors then claim that nuclear shape indicates "biocompatibility". Biocompatibility is a very vague term which is context dependent (see Williams DF. *Biomaterials*. 2008 Jul;29(20):2941-53. Or Bryers JD, Giachelli CM, Ratner BD. *Biotechnol Bioeng*. 2012 Aug;109(8):1898-911.). If the authors mean to demonstrate cellular viability, why do they not utilize a live/dead stain, or similar. The results in A and B are non-quantitative. It would seem that they could easily be quantified. In C, why are cytokines quantified by flow cytometry instead of ELISA/Luminex or similar? The authors claim many fold enhancement of cytokin secretion over LPS stimulated controls, but the responses in these panels looks nearly identical to me. Again, is this quantifiable, with a statistical measure? No statistical significance is indicated in figure 1d.

In figure 2b, the authors demonstrate a slight increase in membrane leakage as compared to a negative control. How do nGO-PEG compare to nGO? do nGO promote leakage? With regard to the migration of macrophages, the authors note that nGO-PEG promote trajectories that approached the edge of the culture plates. What does this suggest? Why would cells behave this way, and what are they migrating towards/away from? The trajectory analysis was based upon only 6 cells. How many times were the experiments repeated? Can a quantitative measure be used to say that this migration is statistically different than the control? Again, what are the effects of nGO in this assay. Comparison against control shows differences, but one would assume that comparison against nGO or a non-2D nanoparticle would provide more robust evidence.

Similar issues of controls and statistical significance remain an issue throughout the remaining figures of the manuscript

In terms of the conclusions, most appear to be supported, but I do not see proof that the interaction of nGO-PEG and cell membranes is via mechanical stimuli. The effects do appear to be mediated through integrin $\beta 8$, but this alone does not prove that the effects are mechanical.

The manuscript would be significantly improved with comparisons to nGO throughout, and possibly to a non-2D nanoparticle given the differences the authors see and claim between 2D passivated and non-2D passivated surfaces.

Reviewers' comments:

Reviewer #1 (Remarks to the Author):

The authors in this article present some new evidences about the role of PEG functionalization of GO as a passivation method in biomedical applications. In contrary with what is believed up to now, it is showed that pegylation of GO induces the highest level of cytokine secretion, which means analogous macrophage activation. The article is well-documented and written and present very important and novel results and I would suggest the publication to 'Nature Communication' after a minor revision according to the following comments.

Below some comments related the manuscript.

a) The phrase in line 103 seems to be in contrast with Fig. 1a. (absence of intercellular fluorescence).

Response: We apologize for any confusion related to the wording of that sentence – we meant to convey that, while the signal is obvious in the case of pristine nGO, no signal can be seen for nGO-PEG. This sentence has been expanded to read

Negligible internalization of nGO-PEG by macrophages was indeed observed in our experiments, as indicated by the absence of intracellular fluorescence signal (purple) in the nGO-PEG image (Fig. 1a). Signal from internalized nanosheets is clearly present after exposure to pristine nGO.

b) The phrase in line 150, 'becomes more hydrophobic after the nGO-PEGs adsorption'; how this is happen since PEG is highly hydrophilic?

Response: We appreciate our reviewer's careful scrutiny of the text. It was meant to be more "hydrophilic", but somehow it was replaced with "hydrophobic" in typographical error; the error has since been removed (in fact, that entire passage has now been reconfigured).

c) The authors should mention the origin and the characteristics of GO (single layers, few layer nanosheets etc). The size of GO nanosheets is very important in bioapplication and it is not clearly referred in the article. Even if we suppose that the mean size of GO is 200 nm (as the bare in the TEM image) it would be very interesting to study GO-PEG with smaller size e.g. 50 nm.

Response: A good suggestion. In fact, detailed characterizations of these same nGOs and nGO-PEGs were included in our previous paper (Reference 13: Nana Luo, et al., ACS Applied Materials & Interfaces (2015): 5239-5247). According to the AFM parameters shown in Figure 1' (attached below) and Figure 3a in the main text, the pristine nGO was 1 – 1.5 nm in height (suggesting a single layer architecture) and ~ 200 nm in lateral size. After chemical conjugation, homogeneous PEG decoration increased the height of the resulting nGO-PEG to ~4-6 nm. To clarify that these data are presented in the literature, we have added the descriptions of "single-layered with an ~200 nm lateral size" and "More detailed characterizations of nGO & nGO-PEG can be seen in the SI (Figs. S1, S2, Table S1) and in previous work.¹³" (**Materials and Methods, Experimental Protocol**, paragraph 1, line 4/line 10).

Figure 1’. Surface morphology of nGO and nGO-PEG and their corresponding height by AFM. Cited from our previously published paper (ACS Applied Materials & Interfaces (2015): 5239-5247).

We also have studied the effects of nGOs within the 50-100 nm size range, finding that these smaller nGO-PEGs induced a level of cytokine secretion comparable to that of the 200 nm nanosheets (see Fig. 2’). We can add this result to the SI if the reviewer feels it necessary.

Figure 2’. Cytokine induction by nGO-PEGs of smaller size (50-100 nm). Cytokine levels were similar to those seen with ~200 nm nGO-PEG.

Reviewer #2 (Remarks to the Author):

1. The manuscript entitled "PEGylated Graphene Oxide Elicits Strong Immunological Response Despite Surface Passivation" describes very detailed insight into the interaction between 2D material and the biological interface (macrophage surface). The detailed molecular biology of an effect of PEGylated graphene oxides on macrophages is further supported by molecular dynamic simulations of interactions between the biological membrane and the tested material. However, the Nature family journals require very precise and detailed work in all areas related to the manuscript. The present story totally fails in materials characterization. The basic and standard microscopic (TEM/SEM) and spectroscopic analyses (IR/Raman/AAS/XPS etc.) are missing. This fact disables to understand/explain objectively the phenomena behind the triggering of immunological response of PEGylated GO. For this reason, I recommend to reject the manuscript in its current form. I suggest the authors to address some issues for further submission(s).

Response: We apologize for the lack of material characterization included with our initial submission. However, most such characterization data (AFM/FTIR/Raman) for these same nGO/nGO-PEG systems were included in a recently published paper (ACS Applied Materials & Interfaces (2015): 5239-524), and thus were intended to be cited in the present context. As shown in the AFM image below (Figure 3'), the nGOs and nGO-PEGs used in this work were both ~200 nm in lateral size; pristine nGOs featured flat surfaces with heights of ~1-1.5 nm, while PEG decoration increased the that average height to ~4-6 nm. FTIR spectra were also applied to compare pristine nGO and nGO-PEG: dominant peaks at ~2850 cm⁻¹ and ~1100 cm⁻¹ observed in nGO-PEG spectra are signatures of C-H and C-O bonds (blue circles) derived from

PEG, indicating the successful conjugation of PEG chains to the nGOs. Raman mappings and corresponding spectra for internalized nGO and nGO-PEGs were also collected (the data for 10 $\mu\text{g/mL}$ nGO and nGO-PEGs are highlighted with red rectangles and arrows in Fig. 4'). The green Raman mappings (left) were imaged according to the background-subtracted G band ($\sim 1600\text{ cm}^{-1}$) intensity. Raman spectra were sampled from the boxed areas for each group; all such spectra possessed the characteristic bands (D, G, 2D) of GO. Compared to nGO, nGO-PEG was hardly detected inside macrophages, in a manner consistent with the confocal images presented in Fig. 1a. To clarify where one might find these characterization data, we have added the descriptions of “single-layered with an $\sim 200\text{ nm}$ lateral size” and “More detailed characterizations of nGO & nGO-PEG can be seen in the SI (Fig. S1, S2, Table S1) and in previous work.¹³” (**Materials and Methods, Experimental Protocol, paragraph 1, line 4/line 10**).

Figure 3^o. AFM characterization (a) and FTIR spectra (b) of nGO and nGO-PEG. Cited from ACS Applied Materials & Interfaces (2015): 5239-524.

Figure 4^o. Intracellular Raman mappings and corresponding spectra of internalized nGO complexes at the concentration of 10 and 40 µg/mL. Citing ACS Applied Materials & Interfaces (2015): 5239-5247)

In addition to the above data, we have also added zeta potential, XPS, TG and AAS results, according to our reviewers' suggestions. Detailed descriptions of these results are included below.

2. Zeta potential measurements for tested nGO and nGO-PEG are missing. The surface charge is an important parameter affecting the interactions of nanomaterials on the nano-bio interface. Is there any change of this parameter in the mixture of nGO/PEG which was tested for cytokine secretion?

Response: All good points and suggestions. In preparing this revision, we conducted zeta potential measurements on both nGO and nGO-PEG, which yielded results in line with those presented in previous work (Reference 13: Nana Luo, et al., ACS Applied Materials & Interfaces (2015): 5239-5247). Based on the pristine nGO potential (-34.01 ± 1.50 mV), the chemical conjugation of PEG screened negative charges and nearly elicited electrical neutrality (-7.89 ± 1.67 mV) on the nGO surface. This change in surface charge would directly influence interactions between nanomaterials and cell membranes, helping to generate different internalization and biological outcomes. We have added these zeta potential data in Table S2.

We also thank our reviewer for suggesting that we obtain a zeta potential for the nGO/PEG mixture. These measurements yielded a zeta potential of $\sim 26 \pm 2.50$ mV (data are means \pm SD with $n=3$), indicating that weak PEG adsorption onto nGO reduces the effective surface charge to a small extent. The screening effects associated with the unconjugated PEG group, however, are still much less dramatic than those seen with conjugated PEG chains.

3. The given AFM of nGO-PEG does not contain the scale bar, similarly as for the carbon particles. A more detailed microscopic characterization should be added, e.g. the difference in the AFM height-profiles of bare nGO and nGO-PEG can support the theoretical model of nGO-PEG. The thickness of PEG shell would also be quantifiable.

Response: The small images corresponding to each of the carbon materials were indeed difficult to scrutinize closely. In actuality, the scale bar included on the CNT image (200 nm) was intended to be applied to all three images. To mitigate this source of confusion, we have magnified the images and added the description “Scale bars: 200 nm” to the caption.

We also investigated relevant height parameters for nGO and nGO-PEG by AFM (Figure 3') and compared those values to our computational model. In our simulated configurations, a single layer of adsorbed PEG adds approximately (and slightly more than) 0.75 nm of thickness to each side of the nanosheet surface, meaning PEGylation results in a 1.5 – 2.0 nm increase in the total thickness of *in silico* nGO. Based on the AFM height profiles (1.0 – 1.5 nm pristine; 4 – 6 nm PEGylated), PEGylation results in a 2.5 nm – 5 nm increase in nGO thickness experimentally. The simulated system, therefore, exists near the bottom of the experimental range. This slight deficit in the thickness of the computational model is expected, given that we had to use relatively short PEG chains and small nGOs to keep our simulations tractable. The additional thickness of experimental nGO-PEGs with longer PEG chains also suggests that dense/multilayered PEG assemblies might form on these nGO surfaces, which we expect to further enhance the phenomena observed in our simulations (i.e., inhibition of membrane insertion and strong face-on membrane interactions), because these dense/multilayered PEG assemblies should be even more capable of excluding edge-on penetration and facilitating face-on binding.

4. The authors show the cytokine secretion as the function of surface area of carbon particles, carbon nanotubes and graphene oxide sheets. How did they measure this parameter (surface area)? This information should be given in ESI.

Response: We have added the related parameters and corresponding formulas in the SI (**Synthesis and characterization of pegylated carbon materials**, last paragraph). Surface areas were calculated using corresponding ideal formulas and approximate estimates of the linear parameters on which they depend:

The approximate surface areas of CNTs, CSs and nGOs were calculated using the following equations: $S_{\text{CNT}} = \pi \times d \times l$ ($d = 20$ nm, $l = 4$ μm), $S_{\text{CS}} = 4\pi \times r^2$ ($r = 100$ nm), $S_{\text{nGO}} = 2 \times d^2$ ($d = 200$ nm,

conceived as a square). The resultant ratios of surface areas were thus $S_{\text{CNT}}: S_{\text{CS}}: S_{\text{nGO}}=3.14: 1.57: 1$; we used these ratios to normalize nanomaterial doses among the different carbon materials.

5. The effectivity of the PEG attachment (the total coverage) of graphene oxide sheets has to be determined because it is the crucial point in further biological experiments. For example, XPS analysis with determination of nitrogen (from NH_2 -PEG) should give information on the amount of PEG chains per particle. Similarly, a TG analysis should be involved.

Response: Another good suggestion. To address this point, we performed XPS, TG, and elemental analysis experiments and condensed the results into Figs. S1 and S2. As Figure 5' (S1) illustrates, the typical C1s XPS spectra for nGO and nGO-PEG were deconvoluted to gauge the relative abundance of important functional groups. After deconvolution, the peak 2/peak 1 area ratio increased from 1.335 to 5.452 from nGO to nGO-PEG, a difference that largely originates from the abundant C-O bonds in PEG. Considering the near-neutral zeta potential of nGO-PEG, as well, our data agree that PEG chains were successfully conjugated onto GO surfaces in our experiments.

Figure 5' (S1). Typical XPS C1s spectra of nGO and nGO-PEG

As XPS can only offer semi-quantitative results (Yao Chen et al, Carbon (2011): 573–580; Guangsheng Tang et al, Nanoscale (2013): 422-428), we also applied thermogravimetry (TG) and elemental analysis to estimate the PEG content of nGO-PEGs. The red curve in the DTG plot (Figure 6') demonstrates the ranges of decomposition temperatures for nGO and PEG, results in line with literature values (Hung-Wei Yang et al, Biomaterials (2013): 7204-7214; Bo Li et al, International Journal of Nanomedicine (2014): 4697-4707). Combining these observations with TG curve measurements, we found the nGO component experienced a clear weight loss (decreasing from 87.9% to 49.7%) between $\sim 100^{\circ}\text{C}$ and $\sim 300^{\circ}\text{C}$. The thermal decomposition of the PEG component followed the decomposition of the nGO component above $\sim 300^{\circ}\text{C}$ (where the weight changed from 49.7% to 36.6%). The above-calculated weight proportions of nGO (38.2%) and PEG (13.1%) suggest an nGO:PEG mass ratio of approximately 3:1. The elemental analysis indicates the carbon mass component in nGO-PEG of about 40%. As 5 nm*5 nm graphene sheets contain about 1000 carbons, the number of PEG (X) was calculated using the simple relation $(1000*12/0.4)/(2000*X)=3:1$. Thus, the average number of PEG chains conjugated to an ideal 5 nm x 5 nm nGO was about five, in close agreement with the six chains conjugated to the nGO in our *in silico* model system. Since the PEG chains represented in simulations were somewhat shorter than those used in experiments, PEG monomer densities are in even better agreement between simulation and experiment. The corresponding data have been added to Fig. S2 and related descriptions have been added in the experimental section of the SI (**Synthesis and characterization of pegylated carbon materials**; paragraph 4, line 2).

Figure 6' (S2). Thermal analysis (TG) and differential thermal gravity (DTG) curves for nGO-PEG.

6. Some quantification of metal impurities should be added to exclude their possible role in the biological response (typically AAS).

Response: As reported in the literature (A. Primo et al, Nature Communications (2014), 5:5291; F. J. Tolle et al, Carbon (2014): 432-442), the heavy metals Mn (major), Cu (minor), and Fe (minor) are sometimes involved in the preparation of GO. Indeed, we noticed the potential impacts of these heavy metals in some of our previous work (Hua Yue et al, Biomaterials (2012): 4013-4021): cell activity was greatly compromised when using a GO sample not subjected to heavy metal removal (Figure 7'). However, we determined that extensive washing could protect these cells from heavy metal damage. Guided by these observations, we also applied protective pretreatments in this work. To remove heavy metals from GO, a 3% H₂O₂ solution was first employed to reduce residual KMnO₄ and MnO₂. The solid product was then separated by filtration, washed repeatedly with 5% HCl solution until sulfate could no longer be detected with BaCl₂, and finally washed with deionized water to neutrality. This extensive washing procedure allows us to carry out *in vitro* experiments in the presence of relatively benign nGOs. The detailed process of removing the metal in our

GO has been summarized in the SI (**Synthesis and characterization of pegylated carbon materials**).

Based on our reviewer's suggestion, we measured the heavy metal content before and after our washing procedure. AAS data (Table 1') show that all heavy metals were greatly depleted upon washing, ruling out possible effects of metal impurities on observed biological responses. The corresponding data can also be found in Table S1.

Figure 7. Cytotoxicity assays of graphene oxide (GO) in macrophages and non-phagocytic cell lines after 48 h incubation. Cited from our previous work (Biomaterials (2012): 4013-4021).

Table 1' (S1) The heavy metal content of GO before/after washing

wt%	Mn	Cu	Fe
Before washing	1.1%	0.02%	0.4%
After washing	0.01%	ND	ND

7. In the confocal images, the nGO-PEG should be monitored by purple/violet color. Does the sample exhibit any fluorescence? Why the fluorescent spectra are not involved? What is the quantum yield? Hence, the signal from nGO is not visible in Fig 1a /the second picture. If there is any signal, the authors should mark it by arrows.

Response: Taking advantage of GO's intrinsic photoluminescence, the intracellular nGO complexes were monitored at the PE-Cy7 wavelength during confocal imaging and marked with false purple in Fig. 1a. Similar methods have been reported throughout the literature (Xiaoming Sun et al, Nano Res., 2008: 203-212; Yin Zhang et al, Nanoscale, 2012: 3833-3842). And though we, too, would be interested in measuring the quantum yield of nGO-PEG, we have yet to find a reliable method by which it can be quantified. In general, literature consensus about the quantum yield of nGO-PEG is lacking. Sun (Nano Res., 2008: 203-212) mentioned that the difficulty in measuring the fluorescence quantum yield of nGO-PEG might originate from the heterogeneous nature of the complexes. Cell imaging, however, can be practically based on nGO-PEG's intrinsic fluorescence in the NIR region, where the auto-fluorescence background of biological tissues is extremely low (Kai Yang et al, Chem. Soc. Rev., 2013: 530-547; Kevin Welsher et al, Nature Nanotechnol., 2009: 773-780).

We also apologize for the poor initial quality of Figure 1a. To clearly present the fluorescence signal associated with pristine nGO, we have magnified the internalization images and marked that signal with white arrows. Images related to nuclear shape have been relegated to the SI (Fig. S3).

8. In the simulation setup, harmonic restraint should be described in more details, e.g. how many atoms are restrained (the force constant of restraint etc.).

Response: A good point. We have added the following text to the SI to clarify how the harmonic restraints were applied in our simulations:

More specifically, a small scaled force constant of 10 kcal/mol/Å² was applied to all graphene sheet carbon positions (including those of oxidation sites and those of carbons directly bound to PEG chains), yielding a total of 1,008 restrained atoms.

9. Simulations of edge-on nGO with or without PEG are not completely equilibrated - there is still a trend in Figure 4.

Response: We agree that sufficient convergence in these simulations is a very legitimate concern. In light of this comment, we extended our insertion simulations by greater than a factor of two, to a full microsecond in length.

As the revised Fig. 4 (Fig. 8') illustrates, the COM separations between the GO nanosheets changed little over the added simulation time. These data suggest that, for practical purposes, the insertion process has converged to a state in which the pristine nGO has (almost) completely inserted into the lipid bilayer and in which ~1 nm of the nGO-PEG remains exposed above the membrane surface. The precise COM separation values continue to fluctuate, of course, as they must within a temperature-constrained ensemble. The data, however, do suggest they converge well over the period of our extensive molecular dynamics simulations.

The most important conclusion that can be condensed from these insertion simulations, we would argue, is that pristine nGOs appear to penetrate cell membranes much more readily than do nGO-PEGs. While both nGOs and nGO-PEGs can likely be internalized via passive mechanisms, the much thicker nGO-PEGs caused by adsorbed PEG chains suggest that passive membrane translocation is much more unlikely with nGO-PEGs than with pristine nGOs. As a direct consequence, one would expect both the internalization of nGO-PEGs and damage to

nGO-PEG-exposed membranes to be lower than in corresponding pristine nGO cases. These expectations are confirmed by our experimental data (Fig. 13' below).

Fig. 8'. Simulations of bare and PEGylated nGO-membrane interactions, initiated from edge-on nanosheet configurations. Left: System snapshots relevant to the observed membrane insertion processes, with GO carbons represented in gray and covalently linked PEG chains rendered in purple. Right: COM displacement data recorded over the course of the simulation trajectories. The PEG/GO COM separation trace (which highlights the PEG extrusion process during insertion) is presented on an alternative vertical scale.

For completeness, we also extended our simulations of nGOs/nGO-PEGs in face-on configurations by a factor of 3, reaching a total of 600 ns in length. While the results of our extended simulations are very much consistent with those presented in the original submission, an additional insight can be gleaned from these new data: after sufficient time interval, PEG loops and termini desorb to form effective “anchors” into the cell membrane (Fig. 9'). As discussed in the revised manuscript (**Molecular Basis of Membrane-nGO-PEG Interactions**, paragraph 4), these anchors further augment the superior binding energy between nGO-PEGs and the membrane surface (in relation to bare nGOs). This observation shines more light on the downstream effects this stronger adhesion might have in terms of macrophage activation.

Fig. 9. Simulations of bare and PEGylated nGO-membrane interactions, initiated from side-on nanosheet configurations. Left: Representative snapshots of the membrane adsorption process. Right, top: total interaction energies between bare and PEGylated nGOs and the lipid bilayer. Right, bottom: illustration of PEG desorption events that lead to the burial of PEG anchors in the membrane, events that further enhance the nGO-PEG/membrane binding energy.

10. Discussed lipid extraction or effect on membrane curvature was not observed in any simulation?? Also the changes in the dynamics of the membrane should be calculated from the simulations (at least for the equilibrated side-on simulations) prior to any discussion of the significance of the computation data.

Response: Lipid extraction effects were indeed observed in the simulations (particularly in the pristine nGO simulation), as noted in the original SI document (**Supplementary discussion for pristine nGO simulations**).

Unfortunately, observing negative membrane curvature of the type proposed in the literature and mentioned here is beyond the scope of our atomistic MD simulations. One would expect that the curvature needed to recruit membrane-integral proteins would occur on the scale of tens to hundreds of nanometers. To allow for computational tractability, the extent of the membrane patch (~7.5 nm x 7.5 nm, after equilibration) used in our simulations was little larger in breadth than the side-on nanosheets themselves (5 nm x 5 nm), and almost certainly insufficient for capturing the type of long-wavelength membrane deformation referenced in the text. Indeed, as illustrated by the images in Fig. 5, no appreciable membrane curvature is seen over

the course of our simulations.

Fig. 10'. Lipid head group diffusion constant calculations in (top) an isolated membrane bilayer and membranes to which face-on bare (middle) and PEGylated (bottom) nGOs have adsorbed. Though increased macrophage activation gives rise to observations of heightened membrane mobility in our experiments, diffusion in membrane regions directly contacting adsorbed nanosheets is expected to be arrested. Our simulations support this expectation: diffusion slows by a factor of approximately 20 after nanosheet adsorption. This phenomenon echoes reports of glass transitions seen in lipid membranes upon substrate binding. Diffusion constants are calculated as a function of the mean-square deviation over time; to confirm the Markov property in the diffusive dynamics, diffusion constant values are calculated as a function of a lag time window and reported at the threshold of (approximate) invariance with that parameter.

That said, changes in lipid diffusion parameters are certainly measurable in the context of our simulations. Short-range interactions between the nanosheet and membrane should actually pin lipids to the nGO surface, arresting diffusion and inducing a loss of mobility. Our calculations confirm this hypothesis, as lipid

diffusion beneath bound nGO/nGO-PEGs is slowed by a factor of ~ 20 , (Fig. S12/10'). The idea of diffusive arrest in lipids in the neighborhoods of membrane ligands is not new, as similar phenomena have been discussed in the context of membrane proteins and lipid rafts (Simons, K. and W. C. L. Vaz. *Ann. Rev. Biophys. and Biomol. Struct.* 33 (2004): 269-295).

The fact that diffusion is arrested in the neighborhoods of nGOs, however, does not exclude the possibility of accelerated diffusion across entire cells. As we hypothesize in the main text, simply increasing macrophage activity (and, thereby, motility) could very well increase the mobility within the membrane itself; more complex signaling pathways might also be in play. Ascertaining just why lipid diffusion accelerates upon nGO/nGO-PEG binding remains an interesting topic for future study.

This discussion about curvature and diffusion effects is now included in the main text (**Molecular Basis of Membrane-nGO-PEG Interactions**, final paragraph).

11. Without all these data, it is difficult to extract the key message of the manuscript! The role of PEG would be due to so many reasons and the authors should identify the key parameters precisely. Moreover, it should be clearly proved if the PEG effect in stimulation of strong immunological response is generalizable also for other nanomaterials; still, the present data do not prove it.

Response: Aside from nGO, we also conjugated PEG to the 2D nanomaterial MoS₂ (provided by Hua Zhang's Group at Nanyang Technological University) to see whether it could trigger a strong immunological response. As shown in the Figure 11', MoS₂-PEG stimulated a comparable level of cytokines to that of nGO-PEG, suggesting a vital role for planar geometry in cytokine induction. We considered these

data to extend somewhat beyond the scope of the present work; if our reviewer deems it necessary, however, we can add these results to the SI.

Figure 11'. Cytokine induction comparison between 2D materials at a 10 $\mu\text{g/mL}$ concentration.

Reviewer #3 (Remarks to the Author):

The present manuscript, entitled "PEGylated Graphine Oxide Elicits Strong Immunological Responses Despite Surface Passivation", demonstrates that 2D nano-particles can elicit enhanced inflammatory responses even in the presence of an "anti-fouling" coating.

The data and results appear to be novel, and interesting; however, the authors should take the time to more fully explain their results in the context of what is known in the literature. The authors find an unexpected result, but do not do a sufficient job of explaining the potential differences in their findings. Additionally, there appear to be many controls missing which would have made the claims more convincing.

1. There is significant data to support the authors claims; however, some figures are difficult to interpret. The use of bright colors in the figures makes some of them hard to read. Yellow text, for example is not easily distinguished. In figure 6e, the authors state the colocalization of green and red, I cannot see any green in these panels.

Response: We thank our reviewer for bringing these issues to our attention. To provide better visual contrast for the yellow text, we have added a gray background in all implicated legends (Figs. 1c, 1d, 3b, 6b, S3, S4, S11, S17). We also agree that the colocalization signal was difficult to distinguish in the original images. To remedy this problem, we have removed the single-channel images and increased the sizes of the high-magnification insets in Figure 6e. We hope that the yellow spots associated with the colocalization of integrin $\beta 8$ (green) and vinculin (red) are now visible in nGO-PEG image. In the control, virtually no colocalization signal was observed.

2. I have questions about some of the methodology used and how the results are interpreted. In figure 1, the authors state that nGO are internalized, but I cannot see any evidence of this in your figure. The authors then claim that nuclear shape indicates "biocompatibility". Biocompatibility is a very vague term which is context dependent (see Williams DF. Biomaterials. 2008 Jul;29(20):2941-53. Or Bryers JD, Giachelli CM, Ratner BD. Biotechnol Bioeng. 2012 Aug;109(8):1898-911.). If the authors mean to demonstrate cellular viability, why do they not utilize a live/dead stain, or similar. The results in A and B are non-quantitative. It would seem that they could easily be quantified. In C, why are cytokines quantified by flow cytometry instead of ELISA/Luminex or similar? The authors claim many fold enhancement of cytokine secretion over LPS stimulated controls, but the responses in these panels looks nearly identical to me. Again, is this quantifiable, with a statistical measure? No statistical significance is indicated in figure 1d.

Response: nGO internalization is indicated by violet fluorescent spots in the images; to make this fluorescence signal more visible, we have magnified the original images in Figure 1a and marked the violet spots with white arrows.

If materials are cytotoxic to cells, they're bound to affect nuclear characteristics like shape and area. Evaluation via the High Content Screening System is becoming more and more popular as a biocompatibility test (Fabian Zanella, Trends in Biotechnology (2010): 237-245; Edward Jan, ACS Nano (2008): 928–938). We agree, however, that there are many other methods for demonstrating biocompatibility. Indeed, we performed CCK-8, Live-Dead, and Annexin-V/PI assays in previous work (Figure 12') on the same systems. CCK-8 indicated that nGO-PEG was very benign at all concentrations studied and faced negligible endocytosis. The CLSM images derived from the Live-Dead assay exhibit a strong green fluorescence representing live cells in the nGO-PEG group; the red fluorescence signal derived from dead cells was much denser in the nGO group. Furthermore, the nGO-PEG group yielded a nearly double negative result in the Annexin-V/PI assay, illustrating the health of treated macrophages. To clarify these points in the present work, we have added the sentence of “results of other viability tests (such as the CCK-8, Live-Dead, and Annexin-V/PI assays) can be found in our past work¹³” to the main text. (**Results and Discussion, last sentence of paragraph 1**).

Figure 12’. Cell viability detected by CCK-8 (a), CLSM images stained by a LIVE/DEAD assay (b), and Annexin-V/PI (c). Cited from ACS Applied Materials & Interfaces (2015): 5239-5247.

With regard to cytokine detection, the BD™ Cytometric Bead Array (CBA) Mouse Inflammation Kit contains all necessary information about the anti/pro-inflammatory cytokines of interest, with fewer sample dilutions and less time required for high throughput analyses (compared to conventional ELISA). The CBA works on the same principle as a Luminex assay, taking advantage of discrete fluorescence intensities on capture beads. By plotting the standard curve, cytokine concentrations can also be quantified (Fig. 1c). This particular assay has been used in many other articles (e.g., Rosario Jiménez, Cytokine (2005): 45-50; Lydie Trautmann, Nature Medicine (2006): 1198-1202).

Concerning the statement of many fold enhancements over LPS group, the original expression was only meant to apply to the nGO and control groups. We have deleted the phrase of “with many-fold enhancement” to clarify this point.

In Fig. 1c, each cytokine column represents mean value of three replicas – we’ve added a statement to reflect this fact in the caption. We find that error bars make this

kind of bar plot difficult to read; representative error bars are included in Fig. 1d. We can, however, add error bars to Fig. 1c if our reviewer insists.

3. In figure 2b, the authors demonstrate a slight increase in membrane leakage as compared to a negative control. How do nGO-PEG compare to nGO? Do nGO promote leakage? With regard to the migration of macrophages, the authors note that nGO-PEG promote trajectories that approached the edge of the culture plates. What does this suggest? Why would cells behave this way, and what are they migrating towards/away from? The trajectory analysis was based upon only 6 cells. How many times were the experiments repeated? Can a quantitative measure be used to say that this migration is statistically different than the control? Again, what are the effects of nGO in this assay. Comparison against control shows differences, but one would assume that comparison against nGO or a non-2D nanoparticle would provide more robust evidence. Similar issues of controls and statistical significance remain an issue throughout the remaining figures of the manuscript.

Response: We appreciate our reviewer's suggestions – adding an nGO group as a further control indeed made our results more convincing. Though far below the positive control, the addition of nGO did cause statistically significant membrane leakage as compared to the negative control and nGO-PEG. By contrast, leakage results for nGO-PEG and the negative control are effectively within error. These results complement our simulation results suggesting that nGO causes more damage to cell membranes than nGO-PEG.

Figure 13' (2b). Membrane integrity study using LDH leakage assay.

With respect to cell migration, observation areas were randomly chosen and the trajectory movies were captured within one field of view. Therefore, the so-called “edge” refers to the edge of the observation field and not the edge of the culture plate. Starting points for cellular motion were also not the same in any of the raw trajectory images; trajectory traces were aligned to a common origin using the Ultraview analysis system. We have performed similar runs at least 3 times for each experimental group; in the main text, we simply showed representative examples for illustrative purposes. Even based on six cells for each experimental group, the results are still certainly different at a high level of statistical significance (as now presented in Fig. S7). Cell trajectory lengths were quantitatively measured based on a single coordinate axis scaled to $\pm 30 \mu\text{m}$, ultimately defining the $30 \mu\text{m}$ sphere rendered in Fig. 2. To clarify all of these points, we have added the sentence of “The trajectory images were quantitatively measured using a $30 \mu\text{m}$ sphere reference, and individual trajectories were centered using the Ultraview analysis system” to the main text and SI. (page 4, paragraph 2, Figure S6 and S7; **Tracking cell trajectories**).

Figure 14' (2c, 2d). Kinetics of macrophage membrane FRAP (above) and trajectories of cells (below) in the presence in the absence/presence of nGO and nGO-PEG.

Based on our reviewer's advice, we have also added FRAP analysis and trajectory traces corresponding to the nGO group (Figure 11'). Compared to the control, the addition of nGO did accelerate the recovery half time ($t_{1/2} \sim 7.51 \text{ s}$), but to a lesser extent than did nGO-PEG ($\sim 5.5 \text{ s}$). Trajectories of nGO-treated cells were also significantly shorter than those of cells exposed to nGO-PEG.

Figure 15' (S6). Kinetics of macrophage membrane FRAP by CS-PEG and CNT-PEG under similar constraints on surface area and dose ($10 \mu\text{g}/\text{mL}$).

1D CNTs and 0D CSs have also been subjected to FRAP analysis at normalized surface areas and concentrations. As Fig. 15' demonstrates, 2D nGO-PEG promoted the most membrane mobility among the three classes of nanomaterials studied.

According to the above results, we have added nGO-specific data in Figs. 2b-d, rearranged Fig. 2, and added descriptive text for nGO in the main body of the manuscript (**Impact of nGO-PEG on macrophage membranes**, paragraph 2 and paragraph 3). Time-lapsed FRAP images for nGO were incorporated into Figure S5. FRAP data related to CS-PEG and CNT-PEG were placed in Figure S8.

4. In terms of the conclusions, most appear to be supported, but I do not see proof that the interaction of nGO-PEG and cell membranes is via mechanical stimuli. The effects do appear to be mediated through integrin b8, but this alone does not prove that the effects are mechanical.

Response: We agree with the referee. Indeed, there is no sufficient evidence to argue for a mechanical mechanism versus, for instance, a chemical one. Activation of the integrin family is often associated with some mechanical force (E Puklin-Faucher, et al, Journal of Cell Science (2009): 179-186; Paul A. Janmey et al, Annual Review of Biomedical Engineering (2007): 1-34), so we simply postulated that our activation mechanism here could be related to some kind of mechanical stimulus. However, as we have no measurements of any kind of force or stress, we replaced the original expression of “mechanical stimuli” with the more general “signal transduction” throughout the text.

5. The manuscript would be significantly improved with comparisons to nGO throughout, and possibly to a non-2D nanoparticle given the differences the authors see and claim between 2D passivated and non-2D passivated surfaces.

Response: Based on our reviewer’s suggestion, and as noted above, we have added several relevant experiments for nGO and analyzed the disparities between 2D and non-2D nanomaterials.

Reviewer #1 (Remarks to the Author):

The authors have carefully worked on the reviewer comments and the manuscript is greatly improved. I suggest the publication of this article in the present form.

Reviewer #2 (Remarks to the Author):

In the revised version of the manuscript, the authors addressed the majority of issues criticized/suggested by the reviewers. Definitely, the manuscript is considerably improved due to added experimental data and their deeper discussion.

I highly appreciate the new experiments regarding the materials characterization and the comparison with nGO and other 2D materials.

Despite of the above mentioned improvements, I still feel that some additional experiments should be carried out to support the mechanistic aspect of the work and, most importantly, to explain the phenomena behind the strong immunological response triggered by the functionalized GO. As soon as the below mentioned key issues are addressed, the reviewer is ready to consider this work for further processing in Nature Communications.

- Still, the key message of the manuscript is rather cloudy. Certainly, the fact that GO, despite of the extensive PEGylation, induces the immunological response is very important finding. However, the mechanistic explanation offered by the authors, is based on the specific behavior of the 2D surface with the cell. This is supported by a nice comparison with PEGylated carbon spheres and carbon nanotubes. Definitely, this comparison should be done also for native CSs and CNTs without any stabilization. If the authors are right, the same trend should be observed and the specific effect of a 2D surface would be even more pronounced.

- Similarly, the generalization of the phenomenon is always required. From this point of view, the new experiments done with the PEGylated MoS₂ represent a strong support of the principal effect of 2D morphology. Again, the comparison with a native MoS₂ or other 2D materials (e.g. fluorographene) would be useful.

- The newly performed experiments (XPS/TG) showed a high content of PEG in the system. The authors do not address the effect of PEG concentration on the observed immunological response. I believe this concentration dependence should also be addressed.

- In the manuscript, the authors state: " Neither free PEG nor an nGO/PEG blend aroused macrophage activity on the scale seen with nGO-PEG (Fig. S4). Chemical conjugation between PEG and nGO, thus, seems to be important for the marked increase in cytokine secretion observed with PEGylated GO." This is slightly controversial with the statements within the text where a 2D morphology is assigned as a key player responsible for the observed response. What do the authors mean under nGO/PEG blend? What is a difference compared to nGO-PEG system studied in the manuscript? It looks the authors expect some kind of specific (covalent) conjugation between PEG and GO. This point should be clarified precisely.

- The authors describe carbon spheres with the size around 200 nm as a 0D system. This is far from the size limit usually accepted for zero-dimensional systems. Real carbon dots with sizes of a few nanometers should be studied to assess their immunological response, mainly if the dimensionality is mentioned as one of the key parameters behind the triggered immunological response (!).

Minor issue:

Fig. S2. Thermal analysis (TG) and differential thermal gravity (DTG). It should be corrected: Thermal gravimetry (TG) and differential thermal gravimetry (DTG).

Reviewer #3 (Remarks to the Author):

The revised manuscript submitted by the authors has addressed all of the concerns previously raised. The additional data presented and reworking of the manuscript has improved the interpretation of the work.

Reviewer Comments

Reviewer #2 (Remarks to the Author):

In the revised version of the manuscript, the authors addressed the majority of issues criticized/suggested by the reviewers. Definitely, the manuscript is considerably improved due to added experimental data and their deeper discussion. I highly appreciate the new experiments regarding the materials characterization and the comparison with nGO and other 2D materials.

Despite of the above-mentioned improvements, I still feel that some additional experiments should be carried out to support the mechanistic aspect of the work and, most importantly, to explain the phenomena behind the strong immunological response triggered by the functionalized GO. As soon as the below mentioned key issues are addressed, the reviewer is ready to consider this work for further processing in Nature Communications.

Still, the key message of the manuscript is rather cloudy. Certainly, the fact that GO, despite of the extensive PEGylation, induces the immunological response is very important finding. However, the mechanistic explanation offered by the authors, is based on the specific behavior of the 2D surface with the cell. This is supported by a nice comparison with PEGylated carbon spheres and carbon

nanotubes. Definitely, this comparison should be done also for native CSs and CNTs without any stabilization. If the authors are right, the same trend should be observed and the specific effect of a 2D surface would be even more pronounced.

Response: In support of our reviewer's suggestion, we have collected cytokine induction data corresponding to pristine CS, CNT and GO under both equal surface area and equal concentration (10 µg/mL) constraints. Under such conditions, these carbon materials were negligibly cytotoxic and were internalized only to a small extent. The cytokine levels induced by each group, though uniformly lower than those related to their PEGylated analogues, followed the same trend as that seen in Fig. 3 in the main text (see Fig. 1' below). Significantly, pristine 2D nGO induced the highest level of cytokine secretion among the materials studied, corroborating the importance of planar structures in elevated cytokine induction. We have added a line to the main text (**Page 4, Paragraph 3**) and a new supplementary figure (**Fig. S10**; same as **Fig. 1'**) to relay these ideas.

Fig. 1'/Fig. S10. Macrophage cytokine secretion levels upon exposure to various pristine carbon nanomaterials, normalized by both surface area and concentration. Though uniformly diminished in extent, cytokine secretion levels induced by these nanomaterials conform to the trend seen in Fig. 3b. Each segment of the bar plot represents the mean of three replicas.

Similarly, the generalization of the phenomenon is always required. From this point of view, the new experiments done with the PEGylated MoS2 represent a strong support of the principal effect of

2D morphology. Again, the comparison with a native MoS₂ or other 2D materials (e.g. fluorographene) would be useful.

Response: We have collected data related to pristine and PEGylated 2D MoS₂, as requested. As shown in Figure 2', MoS₂ stimulated levels of cytokine secretion very comparable to those induced by nGO in both pristine and PEGylated states. These data further support the notion that the general physicochemical properties of 2D nanomaterials are important at the bio-nano interface. A sentence has been added to the main text (**Page 8, Paragraph 3**) to describe this new supplementary figure (**Fig. S22**).

Fig. 2'/Fig S22. Comparison of cytokine secretion levels between pristine/PEGylated nGO and MoS₂ at 10 µg/mL nanomaterial concentrations. MoS₂ and nGO stimulated comparable levels of cytokine secretion with and without PEGylation.

The newly performed experiments (XPS/TG) showed a high content of PEG in the system. The authors do not address the effect of PEG concentration on the observed immunological response. I believe this concentration dependence should also be addressed.

Response: A good suggestion. To address this point, we prepared two additional classes of nGO-PEGs with moderate degrees of decoration (at 20% and 80% of the original PEG density). Our simulation results suggested that elevated cytokine induction is intimately related to interactions between PEG chains and cell membrane lipids; these observations imply that increasing PEG densities would only extend residence times on the cell membrane and thus elicit an even stronger cytokine response. Our new experimental results confirm this hypothesis, as we found that cytokine secretion was positively correlated with PEG density (Fig. 3'). We have adjusted the corresponding description in the main text (**Page 3, Paragraph 2**) and included these data in **Fig S5**.

Fig. 3'/Fig. S5. Impact of PEGylation density on cytokine induction.

Cytokine secretion increases gradually with increasing PEG densities.

In the manuscript, the authors state: “ Neither free PEG nor an nGO/PEG blend aroused macrophage activity on the scale seen with nGO-PEG (Fig. S4). Chemical conjugation between PEG and nGO, thus, seems to be important for the marked increase in cytokine secretion observed with PEGylated GO.” This is slightly controversial with the statements within the text where a 2D morphology is assigned as a key player responsible for the observed response. What do the authors mean under nGO/PEG blend? What is a difference compared to nGO-PEG system studied in the manuscript? It looks the authors

expect some kind of specific (covalent) conjugation between PEG and GO. This point should be clarified precisely.

Response: We are sorry for our probably not-clear-enough description with respect to the nGO/PEG blend used in our experiments. Indeed, this nGO/PEG blend was a simple combination of nGO and PEG that was not subjected to (covalent) conjugation; such conjugation is achieved in the nGO-PEG system using a non-trivial chemical process (see **Materials and Methods**). To clarify this point, we have replaced the former “nGO/PEG blend” statement with “a simple mixture of nGO and PEG (featuring no covalent conjugation between the two groups)” in the manuscript. Furthermore, we have changed the words “nGO/PEG blend” to “nGO+PEG mixture” in the **Fig. S4 caption**, and we have replaced the final sentence of that caption with, “Therefore, both 2D structure and the chemical conjugation of PEG seem essential for the achieving high levels of cytokine secretion.”

The authors describe carbon spheres with the size around 200 nm as a 0D system. This is far from the size limit usually accepted for zero-dimensional systems. Real carbon dots with sizes of a few nanometers should be studied to assess their immunological response, mainly if the dimensionality is mentioned as one of the key parameters behind the triggered immunological response.

Response: We thank our reviewer for bringing this point to our attention. As a matter of fact, the extremely small size of true 0D materials presents a hurdle for PEGylation (as the PEG chains used in this work are much larger than fullerenes themselves), and may also result in the emergence of alternative internalization pathways. Therefore, carbon dots, in our opinion, are not suitable for comparison with the materials studied in this work. We do agree that the size of the carbon spheres in our experiments (~200 nm) is larger than that of typical classic 0D nanomaterials. To avoid any misunderstanding, we have changed our descriptions of these materials to simply read “carbon spheres” throughout the manuscript.

Minor issue:

Fig. S2. Thermal analysis (TG) and differential thermal gravity (DTG). It should be corrected: Thermal gravimetry (TG) and differential thermal gravimetry (DTG).

Response: We have amended the above sentence, as requested.